# Land and Water Usage in Beef Production Systems

**DOI:** 10.3390/ani9060286

**Published:** 2019-05-28

**Authors:** Donald M. Broom

**Affiliations:** St Catharine’s College and Department of Veterinary Medicine, University of Cambridge, Madingley Road, Cambridge CB3 0ES, UK; dmb16@cam.ac.uk

**Keywords:** sustainability, land resource usage, water usage, beef production, silvopastoral

## Abstract

**Simple Summary:**

Consumers increasingly demand sustainable food production, including using world resources efficiently, avoiding environmental damage and ensuring good welfare of animals. Reports have suggested that beef production is costly in relation to world resource use and greenhouse gas production, so some consumers avoid beef. However, many reports refer mainly to feedlot systems. Ruminants can eat leaves that humans cannot eat, so if they are not fed grain, systems can be sustainable and valuable. This paper presents an analysis of the production of beef comparing all aspects of the use of land and conserved water for four production systems. It is suggested that conserved water use is a useful measure. Land use was the highest in extensive unmodified pasture systems, especially if the land became degraded. Less land was used in both feedlot and fertilised pasture systems and much less in semi-intensive silvopastoral systems. Conserved water use was the highest in feedlot systems, partly because of the grain fed to the cattle, lower in pasture systems and lowest in semi-intensive silvopastoral systems. This research indicates that, when beef production systems are being selected or consumers are deciding which beef to buy, extensive systems that degrade the land should be avoided, and well-managed extensive systems, especially semi-intensive silvopastoral systems, should be preferred to feedlot systems.

**Abstract:**

This analysis, using published data, compared all land and conserved water use in four beef production systems. A widespread feedlot system and fertilised irrigated pasture systems used similar amounts of land. However, extensive unmodified pasture systems used three times more land, and semi-intensive silvopastoral systems used four times less land, so the highest use was 13 times the lowest. The amount of conserved water used was 64% higher in feedlots with relatively intensive rearing systems than in fertilised irrigated pasture; in extensive unmodified pasture systems, it was 38% and in semi-intensive silvopastoral systems, it was 21% of the fertilised irrigated pasture value, so the highest use was eight times the lowest. If there was no irrigation of pasture or of plants used for cattle feed, the feedlot water use was 12% higher than the fertilised pasture use and 57% higher than that in semi-intensive silvopastoral systems. These large effects of systems on resource use indicate the need to consider all systems when referring to the impact of beef or other products on the global environment. Whilst the use of animals as human food should be reduced, herbivorous animals that consume food that humans cannot eat and are kept using sustainable systems are important for the future use of world resources.

## 1. Introduction

Consumers increasingly demand that the production of food, and of other products, should be sustainable and they consider the ethics of production methods when evaluating product quality [1]. A system or procedure is sustainable if it is acceptable now and if its expected future effects are acceptable, in particular in relation to resource availability, consequences of functioning and morality of action [2]. The efficient usage of world resources is becoming a key issue in the assessment of sustainability. For beef production, the total amount of land used and the water footprint [3,4] are considered in this paper, but it is suggested that the total amount of conserved water used is a useful measure. The hypothesis tested is that beef production systems vary in the amount of land and conserved water that has to be used for them to function. The extent of the variation that is calculated here shows the inadequacy of single-figure representations of the impact of beef production in the world. Statistics referring only to feedlot systems are not valid for the most sustainable beef production systems. In order for a food production system to be sustainable, it is important to consider key resources, such as land and water, but also other components of sustainability such as the welfare of the animals used, the welfare of humans and the environmental impacts including those on biodiversity, pollution and world climate [5,6,7]. All aspects should be considered in evaluating sustainability [8], and any aspect can make a system unsustainable.

## 2. Material and Methods

In this study, the usage of land and conserved water (defined below) were estimated for all aspects of four methods of beef production. These included the production of plants that the animals eat directly and imported feed, mainly grain. These production methods were selected because of their importance now, their expected future importance and the availability of data. Most of the data used came from South America and North America, but some came from Australia, FAO, or other world sources, and they were selected because they are typical of world production. Apart from the world data, in each of the datasets for the systems utilised here, the beef produced is a significant part of national production, and some is exported to different parts of the world. This study provides evidence for consumers and others, when they are deciding which beef to buy, about the relative impacts on the use of land and water in four production systems. It is not an attempt to match systems in one area. The species of plant grown varied from area to area, but the four beef production systems could use different plant species in different regions of the world. There was variation among breeds of animals, but the breeds for beef production were those that are normal for the system in the locality from which the data were taken. *Bos taurus*, zebu (*Bos indicus*), or zebu-cross beef breeds were used for the calculations. Data are not available for animals matched for breed where each of the four systems were operated in one place. Indeed, these data would not be meaningful as the breed would not be adapted to each of the conditions. Whilst some of the data used were for comparisons within a geographical area, there were no areas where all four systems were commercially usable, so comparisons within an area would not be valid.

In relation to the use of world water resources, zebu cattle are important because they use approximately 71% of the water consumed by *B. taurus* cattle per unit of food consumption, as they are better adapted to higher temperatures [9]. Water usage is seldom recorded (see below), so the amount of data available is small. As a consequence, it was necessary to use data from many different sources for the components of beef production. The data were related to animals from birth to slaughter, so variation amongst systems in the typical duration of phases of production was taken into account. The information about land and water usage was obtained for a sequence of periods starting at birth, as explained for each of the four systems, and then expressed as a total lifetime figure. Breeding beef cattle were not separate populations from production animals. They had the same role in each system considered, and the breeding animals themselves were used for beef production. All parent animals used in beef cattle production were included in the calculations. Most of the data came from conditions that are subtropical or at least hot in summer, but temperature affects the variables so, where this occurred, the production figures were adjusted to 27 °C. The hot carcass weight figure was used as the cattle production measure throughout this paper.

The first step in the approach and general model for analysis used in this study was to obtain a representative figure for beef production in each system in an area where data for land and water use exist. The second step was to find the density of animals at each stage from birth to slaughter and the land usage and direct water usage in the production of the animals converted to a standard temperature. The third step was to calculate land and water use for food and other resources for animal production in each of the four systems. The area of land and the amount of conserved water per unit of meat production was then calculated for each component of the beef production process. Details of the components of the calculation model are explained further in the sections below. All data used are either quoted here or are in the references cited. The data used came from world calculations by FAO, comparative studies of systems where they exist and are valid, individual studies of systems and studies relevant to the industry as a whole, for example land and water use in a beef processing unit. For some aspects of the analysis, data were scarce, but all of the data used were evaluated as being representative. Although statistical methods were used to obtain most of the data points used, no overall statistical model for the simple calculations conducted was possible. Carefully quantified data for each of the many components and analyses explained below would take many years to collect. However, the results presented are of immediate value for system comparison and are relevant to decisions about which systems are sustainable and should be used in the future.

### 2.1. The Systems

#### 2.1.1. Extensive Unmodified Pasture

A widespread, traditional beef production method is to rear the cattle throughout their lives on pasture, initially with their mothers and then in age-related groups. A widely used system considered here consists in leaving manure from the animals on land which is not treated with artificial fertiliser and is not irrigated. The typical live weight when the zebu or zebu-cross animals are slaughtered at 30 months of age is 468 kg, that becomes 255 kg hot carcass weight [10].

#### 2.1.2. Feedlots

A second system used in many countries, involves zebu or *B. taurus* animals, depending on temperature and vulnerability to tick-borne diseases, being finished in feedlots. The cattle are often kept on fertilised, irrigated pasture when young and then in feedlots at high density with high levels of concentrate feeding for the last few months of life. Some animals are kept extensively prior to the feedlot period, so this situation is also considered in the results table. A typical duration of rearing on fertilised pasture is 16 months, so this duration was used for these animals, but 20 months may be needed to reach the same weight if the animals are on extensive systems, so 20 months was used in these calculations. The site of the feedlots is usually different from that for the earlier rearing, and animals may be transported long distances to the feedlot site. In Mexico, the feedlots are often more than 1000 km from the rearing areas, and the concentrate food is frequently imported from the United States. The data for feedlots correspond to the following time-course: months 0–9 in a suckler herd with the mothers, months 9–16 without the mothers in a paddock and months 16–20 in the feedlot. Some feedlot animals are reared in more confined situations, and some remain in the feedlot for longer or shorter times; the system considered here is frequently used in North and South America. 

If the animals are in a feedlot for the last 4 months, their final weight is higher than if they were on the fertilised pasture throughout. Hence, the figure is based on the common situation of 20 months to slaughter, rather than the normal 30 months on pasture. The final weight used in the calculations was 491 kg live weight, which became 268 kg of hot carcass weight for animals from the feedlot.

#### 2.1.3. Fertilised, Irrigated Pasture

This system is used in many countries, but most of the data used here were from Colombia at altitudes that provide a sub-tropical or warm temperate climate [11,12]. For the data used in this study, the cattle were *B. taurus* breeds, but zebu and zebu-cross were also used at lower altitudes. The irrigation was provided as needed during each year, and fertiliser application was typical for pasture managed by farmers who could afford fertilisers. No supplementary concentrate feed was provided. The animals reached a weight of 468 kg at 30 months of age, similar to those on extensive unmodified pasture, but the density of animals was higher, as food availability was greater.

#### 2.1.4. Semi-Intensive Silvopastoral Systems

Silvopastoral systems with shrubs and trees in addition to pasture plants are used for beef production in Colombia, Brazil, Australia and other countries. The semi-intensive systems utilise a shrub with edible leaves, such as the high protein leguminous shrub *Leucaena leucocephala,* together with trees, some of which also have edible leaves. Such trees are especially valuable in times of low rainfall [7,11,12,13]. The highly palatable, high-protein shrub and tree leaves necessitate the use of rotational management where the cattle are moved from paddock to paddock before they damage the plants, and this allows a higher stocking density to achieve the same growth rate as on the pasture-only systems. The final weight of the animals at 30 months of age was again 468 kg, so this value was used in the calculations. The weight was similar to that of animals on extensive and fertilised pasture, but the density of animals was higher, as food availability and quality were greater. The high protein and high total feed availability obviate the need for artificial fertiliser or supplementary feed. Because of the better soil structure, the higher water-holding capacity and the amount of herbage present, there is little need for irrigation, except in extremely dry conditions. Unless otherwise specified, the data for semi-intensive silvopastoral systems and fertilised, irrigated pasture were from matched studies using the same breeds on the same Colombian farms.

With the exception of the feedlots, the systems analysed here were also used for milk production, with dairy or mixed-use animals, but only beef cattle data are shown here.

### 2.2. Methods for Land Use Assessment

In this comparison, all land used for keeping and feeding the animals was calculated. The area of land used for meat processing was included but was very small [14]. The land used for fertiliser production, farm equipment production, transport of animals, transport of feed and that used by staff working on the farms were also thought to be relatively small and were not included.

#### 2.2.1. Extensive Unmodified Pasture

The land required for beef cattle on extensive unmodified pasture at 27 °C would vary according to water availability and soil type. The figures used here are from FAO [15] and from a study [16] in Colombia in which there was a direct comparison of this system with fertilised irrigated pasture and semi-intensive silvopastoral systems (see also [7]). The data were not from drought conditions nor from degraded pasture, where the land used per unit of beef production would often be much higher. The overall stocking density was 0.5 animals per hectare. For all four systems, the stocking density calculation included parent animals.

#### 2.2.2. Feedlot Systems

The density of beef cattle in feedlots varies, but the typical figure of 140 animals per hectare during the last four months before slaughter was used here. These figures and those for the space required for the rearing with the mothers and the period at pasture before transport to the feedlot were taken from [17,18] and [19]. Each feedlot is normally used for two groups of animals per year. Most of the space required for keeping the feedlot system animals is that during rearing before entering the feedlot. The overall stocking density was 1.5 animals per hectare, including parent animals.

Beef cattle whose production is completed in feedlots are fed a ration composed almost entirely of grain whilst in the feedlot and receive some supplementary feed during earlier rearing. The majority of this feed is maize, but soya and other food sources are also used. Small quantities of concentrate feed fed to beef cattle are by-products of human food, but it was not possible to take account of this. In calculating the land required for this feed production, the maize grain figure was used, as this is the major component of feed. Other feeds, all grown in cultivated fields, are fed to beef cattle; “green maize” production was more kg per ha than maize grain, whilst soya, wheat and barley were less. The maize grain data were considered the most representative. The figure for the amount of land required for maize production was calculated from the world mean maize production of 5755 kg ha^−1^ [20]. Whilst this figure is only half of that obtained in the USA, it is higher than that quoted by FAOstat for most countries in south and central America, e.g., for Mexico it is 3789 kg ha^−1^. The land used to produce 1 kg maize using the world mean is therefore 0.000174 ha. The amount of concentrate fed to an individual animal during earlier rearing and whilst in the feedlot was calculated from data in [19] to be 19.3 kg per kg hot carcass. Hence, the land area per tonne or kg of meat could be calculated.

#### 2.2.3. Fertilised, Irrigated Pasture

Fertilised pasture, often called improved pasture, is used in many parts of the world for beef production. In hotter periods, such as the 27 °C condition considered here, it is often irrigated. The space used from the comparative studies [7,16] is reported here. The pasture was a grass monoculture, of a kind widely regarded as good for beef production, with no supplementary feed provided to the cattle and an overall stocking density of 1.0 animals per hectare.

#### 2.2.4. Semi-Intensive Silvopastoral System

The semi-intensive silvopastoral system provides more nutrients, especially more protein, to cattle than monoculture pasture, so the animals can be at a higher density to achieve the same weight at 30 months of age. In the studies used in this analysis [7,16], the overall stocking density was 3.0 animals per hectare. No supplementary feed is needed.

### 2.3. Methods for Water Use Assessment

The water use calculated here was entirely conserved water. Rain falling on pasture or on fields for crop growing was not included in these calculations unless conserved in human-controlled water storage systems. These included water collected from public and farm reservoirs, human buildings, streams, lakes, wells and aquifers. It is suggested that the figures for conserved water use are more relevant to beef farming and its impact than the total rainfall figures. The amount of conserved water used can be controlled to some extent by farmers, companies and governments, so it is valuable to know the different amounts used in different systems. In many countries, a high proportion of rain runs off rapidly, so it is never used for drinking by cattle or for production of their food. Would the amount of rainfall affect the conserved water data? Drinking by animals in feedlots would scarcely be affected by rain falling in troughs. The amount of water in storage systems is clearly affected by rainfall, so all data presented refer to the lifetime of the animal and, hence, group together periods of greater and lesser rainfall. The amount used for irrigation of pasture would be affected by the amount of rain falling on the land, so a mean figure for several years was used. Some of the water used in beef production is purified for human consumption, and this was noted if the data presented purified water.

The ready availability of relatively cheap water in many countries where detailed studies of animal production systems are carried out has resulted in there being little concern about water usage, except as a relatively small monetary cost. However, much beef production is carried out in countries where water shortage is increasingly common, aquifers are declining in size, and most producers are becoming more aware of the cost of water. In some countries, however, the true cost is not charged to farmers, so some governments are subsidising the beef industry [21]. 

Since water consumption by cattle is substantially affected by ambient temperature [22], figures for each system have been corrected to 27 °C. The widely used [22,23,24] correction of 0.81 litres drunk per day for each degree C increase in temperature is described by [23]. Conserved water used for irrigation of pasture or during cattle food production was included, as was water used in the processing of the hot carcass and for cleaning and manure removal. Water in water courses after waste removal was not included.

#### 2.3.1. Extensive Unmodified Pasture

The data for mean water drunk from birth to slaughter by beef cattle in extensive unmodified pasture were for zebu cattle in Brazil at 27 °C [24]. The calculation per kg hot carcass weight was as described above. In areas where rainfall is higher, the amount drunk at this temperature would be lower. Evaporation from troughs at 27 °C was a similar proportion of total water provided for the four different systems. The measurements used were from [25] and [26]. Trough-washing involves the use of a very small amount of water in the systems where the water is provided in fields, and very little water becomes polluted and has to be removed from the troughs in these systems. Water is not normally administered to the bodies of cattle for cooling in extensive systems, but an unknown amount of water in field reservoirs may be used for cooling by the animals themselves. The use of water in meat processing [18] is included in Table 2 but was assumed to be the same for all systems of beef production.

#### 2.3.2. Feedlot Systems

The data for the amount of water drunk by cattle from birth to slaughter, finishing in feedlots, were from Nebraska [27]. For animals kept extensively and then moved to a feedlot, the water consumption data from extensive systems, corrected to 20 months and for final carcass weight, were used. Water is used for washing troughs, as the high density of animals in feedlots results in frequent contamination of water troughs. At 27 °C, cattle in feedlots are sometimes cooled using water. The data in Table 2 for both washing and cooling are from a Queensland feedlot study [28], as is the loss of polluted water removed from troughs.

Beef cattle in feedlots are fed much more concentrate feed than cattle in other systems, not just when in the feedlot but also during earlier rearing. The data on the irrigation of maize for feed were from FAO [15] for the USA and gave a figure of 21.5 litres of water per kg of maize produced and a total concentrate feed intake by a feedlot animal over its life of 19.3 kg maize per kg hot carcass. Since there are systems in which the production of concentrate feed for beef cattle does not involve irrigation, data for these systems were also included.

The irrigation of pasture during 16 months of pre-feedlot production of beef cattle could be zero but would normally be at the fertilised pasture rate. In the calculation, data relating to 16 months were used instead of those for 30 months of production in that system. A typical amount applied at 27 °C is shown in Table 2 [29] together with figures referring to pasture that was not irrigated during development. 

Water is used for fertiliser production, and a typical fertiliser used for maize and for pasture is urea [30]. The mean maize production in 2006–2016 in Brazil was 40 tonnes per hectare, and the fertiliser to produce this was 100 kg N per ha per year, corresponding to 214 kg urea. For every tonne of urea, 1.54 tonnes of water are needed [31]. Hence, 330 L water were required to produce 40 tonnes of maize, and 0.008 litres per kg maize. Since 19.3 kg maize are used per kg hot carcass, 19.3 × 0.008 = 0.2 litres of water are needed per kg hot carcass.

A typical application of fertiliser on pasture is 200 kg N per ha per year, which is equivalent to 428 kg urea × 1.54 = 659 kg water per ha per year. For the feedlot system of 20-month growth, 16 months were on fertilised pasture, so the land per kg of meat was 0.0052 ha, with 659 kg water × 0.0052 × 17/12 = 4.6 L water per kg hot carcass.

#### 2.3.3. Fertilised Irrigated Pasture

The amount of water drunk by beef cattle on fertilised irrigated pasture was measured in Colombia to be 25% more than that for silvopastoral systems [32]. Pasture irrigation was used in the Colombian fertilised pasture study for which the meat production data are described above [7,16,33]. The amount of irrigation varies from year to year. The duration of the period on the pasture was 30 months, and the amount used for irrigation at 27 °C was based on data from [29].

The application of fertiliser to pasture was calculated as for the feedlot animals during rearing on fertilised pasture, but the land per kg hot carcass weight was 0.01 ha, and there were 30 months of growth, so 659 kg water × 0.01 × 30/12 = 16.5 L water per kg hot carcass weight.

#### 2.3.4. Semi-Intensive Silvopastoral Systems

The amount of water drunk by beef cattle on a semi-intensive silvopastoral system was measured in Argentina [34]. The ratio of this figure to that for extensive, good-quality pasture was similar in a comparison in Colombia. Sheep on silvopastoral systems obtained 19% of their water intake from drinking and the rest from their diet, whereas sheep on fertilised pasture obtained 50% of water from drinking ([35] and Maurício personal communication). The meat production data are described above [7,16,33]. The data presented here are for the typical situation in upland areas of Colombia where animals did not need pasture irrigation. In a comparison across systems, if irrigation were to be needed in the silvopastoral system, the amount of water used for irrigation would increase by approximately the same amount in the irrigated pasture system.

#### 2.3.5. Publication Ethical Statement

No ethical evaluation was needed for the analysis presented.

## 3. Results

The data for land use in the four systems of beef production are presented in Table 1. For Table 1 and Table 2, details including the final weight of animals, duration of growth periods and overall stocking density in the system of production are explained in the Material and Methods Section. These figures include all animals slaughtered for beef production, including the parent animals. Some animals were reared extensively before the period in feedlots, whilst others were reared on fertilised irrigated pasture, so both are shown in Table 1.

The data for water use in the four systems of beef production are presented in Table 2. As in Table 1, feedlot systems with extensive rearing and fertilised, irrigated pasture rearing are shown separately. Separate totals for water use are shown for systems with: (i) irrigation of crops used for concentrate feed and irrigation of pasture, (ii) no irrigation of crops used for concentrate feed, and (iii) no irrigation of crops used for concentrate feed or pasture. The figures in Table 1 are per tonne of meat, whilst those in Table 2 are per kg.

## 4. Discussion

The rising demand for land, both for agriculture and for human dwellings and commerce, has led to shortages of land for these purposes and to great reductions in the land available for wild animals and plants [36,37,38,39]. Even though the land may have a high monetary cost, much of the land use is not very efficient, for example when used for human food production, and land degradation is still increasing as a result of some human practices. It is useful to calculate how much land is used for current food production systems in order to plan how to use the land more efficiently. The calculations carried out here of land use for beef production included all land, that is, not just the land where the animals were kept but also that for producing the food for the animals. 

In agreement with [40], a key result of these calculations of land use per kg of meat produced is that extensive systems with unmodified pasture use most land, corresponding to 2.7–12.3 times the land use of the other systems considered here. If conditions on extensive unmodified pasture are very dry or the land is degraded, even more land is needed to produce a kg of beef. A second result is that the land needed for the feedlot system when the animals are reared on fertilised, irrigated pasture is similar to that needed when all production is from fertilised pasture. If the animals are reared extensively before being put in the feedlot, the land use is twice that for beef cattle kept solely on fertilised pasture. A third result is that the semi-intensive silvopastoral system requires 25–32% of the land needed for the feedlot system, 22% of that for fertilised pasture and 8% of that required for extensive pasture production. The amount of land needed for the semi-intensive silvopastoral system is lower because the combination of pasture plants and nitrogen-fixing shrubs and trees provide all the protein and other nutrients needed by the animals. A careful management of the animals in this system, moving the animals before they damage the plants, is necessary for a more efficient use of the land.

The substantial differences between extensive systems in land use per unit of beef production are important when interpreting results like those of [41], as it is clear that some extensive systems can use less land than some grain-feeding systems. Also, “different livestock systems provide different functions to different human systems and require different strategies, so they cannot readily be pooled together” [42]. These results indicate the need to consider all systems when evaluating the impact of beef or other products.

The quality of the land for agricultural production may not be the same in the four systems, and this will affect land and water use. Extensively grazed pasture may have been of poorer quality for plant growth before any impact of the livestock, feedlots may be on high- or low-quality land, and silvopastoral systems may be initiated on good-quality land or used as a measure to help the recovery of degraded land. The quality of the land after beef has been produced is of particular importance for system sustainability, and soil quality is improved most by silvopastoral systems [43]. The initial state of the land has effects on the costs of running each system, and the systems vary in labour costs. In considering how best to use land, an important factor is that a large proportion of land in the world is suitable for rearing herbivorous animals but not suitable for growing crops. Hence, the best usage, as far as human food production is concerned, is to use optimal systems on it to rear ruminant mammals and other herbivores.

If the amount of land needed for beef production is reduced, more land can be spared for other purposes such as preserving natural habitats, reducing the rate of extinction of living organisms and increasing biodiversity [37,44]. If the beef production system is associated with higher biodiversity and other aspects of sustainability, as it is in silvopastoral systems, there is also land sharing with wildlife, as increasingly demanded by the public.

Permanent, regular or intermittent water shortages in many parts of the world result in efforts to use water more efficiently [45,46]. In the calculations in this study of the water use per kg of beef produced, attempts were made to include all aspects. Hence, conserved water used in producing any food supplied to the animals and in producing fertiliser and other resources, as well as that drunk by the animals, was included. In some temperate areas, there is much water available most of the time, but in many areas where beef animals are kept and where their food was grown, water is in short supply for at least part of each year. Hence, water is a resource that limits the production of plants and animals used for human food. Using the calculations made here, estimates can be made for other similar systems. For example, if a system uses no artificial fertiliser, to calculate water usage, deduct the figure quoted here for fertiliser production.

Comparisons by other authors of grazing and intensive beef production [47] had similar results for the components that equate approximately to the conserved water calculation used here, but the overall result of [48] and other studies was dominated by the very much greater amount of water measured that was rain falling on the land. However, a proportion of this rain could never have been used for agriculture or any other human purpose. That calculation is greatly affected by the area of land used in total production. In the calculations in this paper, only conserved water was considered, and land area results were presented separately. Rain falling on the land was not included, unless it had been captured in a reservoir or other human water-conserving system such as aquifers from which water is pumped. Water taken from streams, rivers and lakes and conserved for human use may or may not limit water availability in other places. Water that is never conserved and which passes down rivers to the sea is of less interest in relation to human activities, unless polluted. Indeed, when very heavy rains occur, especially when these are in limited areas, the rain may have little impact on a pre-existing drought situation and may cause much soil loss and, hence, more water loss in the future. Measures of the amount of rain falling on an area of land would sometimes be closely related to the amount of water that can be conserved and used in plant and animal production. However, at other times, the amount of rain falling would have much less influence on beef production. The amount of conserved water used will always be useful information and of particular value in studies comparing production systems. Hence, these data are novel and provide different information from that presented by most previously published results.

Some conserved water is held and then used to irrigate planted areas or provide water for drinking or other animal production use. Other conserved water is purified so that humans can safely drink it. This purification has monetary and carbon costs, so it is mentioned in the results. The feedlot beef production system generally uses more purified water than the other three, more extensive systems. Comparative information on degradative water use [49] is not available for the four systems. However, it would be expected to be low, except for animals kept at high density, as in feedlots, and for some artificial fertiliser use.

In the extensive unmodified pasture and semi-intensive silvopastoral systems, 89% and 84% of the water used in beef production was drunk by the animals. However, in the feedlot systems where the animals were reared on irrigated pasture and fed concentrates irrigated during production, 12% of the total water used was that drunk by the cattle, while, when fertilised irrigated pasture systems were used throughout, this percentage became 21%. Overall, a widely used feedlot system involves the highest water usage; water usage in fertilised pasture is also high, whereas that in extensive unmodified pasture is substantially lower, and that in the semi-intensive silvopastoral system is the lowest. It is clear from Table 2 that irrigation of crops during concentrate production and during feeding on pasture is the biggest factor affecting total water use. Water use in feedlot systems with no irrigation was only 19% higher than in semi-intensive silvopastoral systems. The estimates [8,40] of water use for grain-feeding of cattle were lower than those calculated in this study but, as mentioned above, the water measurement methods were somewhat different. When considering the drunk water data, in relation to the production of other types of beef, it is relevant that water drunk by zebu cattle was less at 27 °C than that drunk by *B. taurus* cattle, so the figures presented in Table 2 would be higher, perhaps 40% more, in countries where *B. taurus* breeds are used at this temperature.

When other sustainability factors were considered, the four systems differed in some important respects. The welfare of the cattle can be worse because fast growth leads to more leg joint disorders [49], and this is principally a problem in feedlot animals. The feedlot conditions with high stocking density involve the poorest welfare, whereas the animals in the semi-intensive silvopastoral systems have the best welfare [50]. Feedlots may also cause local pollution problems. There are impacts of the systems on greenhouse gas production and some other externalities [51]. If cattle are fed food that humans could eat, such as maize or soya, resources are wasted. The future requires partly the reduction of the consumption of animals as food but also the use of animals fed on a mixture of plants inedible for humans, including some nitrogen-fixing plants with high protein content, so that supplementation and artificial fertilisers are not needed, and less land and water are required.

## 5. Conclusions

The efficiency of land and water use in the production of human food, such as beef, is being questioned. Figures concerning beef production externalities, such as land use and water use, are usually not representative of all beef production systems and may just refer to feedlot systems. Some of the public concludes that beef and other ruminant meat production should cease. However, when ruminants utilise food that humans cannot eat, such as the leaves of herbs, shrubs and trees, the production system can be sustainable and of great value in the world. When land use for all aspects of production of beef, including all sources of food and other materials used, were calculated for four production systems, it was the highest in extensive unmodified pasture, especially if the land became degraded, was 63–67% lower in both feedlot and fertilised irrigated pasture systems and 92% lower in semi-intensive silvopastoral systems. In relation to water use, it is suggested that it is valuable to assess the amount of conserved water used, instead of, or in addition to total water. Conserved water use was the highest in two widely used feedlot systems, 26–39% lower in fertilised irrigated pasture systems, 72–77% lower in extensive unmodified pasture systems and 84–87% lower in semi-intensive silvopastoral systems. Whilst not all systems can be used in all places and there are other components of sustainability, notably animal welfare and greenhouse gas production, information about such usage of resources can be usefully considered.

Implications of this research are that, when governments, food retail companies, or individual farmers are deciding on which beef production systems to use or consumers are deciding on which beef to buy, extensive systems that degrade the land should be avoided, and well-managed extensive systems, especially semi-intensive silvopastoral systems, should be preferred to feedlot systems. Water shortage makes such choices more important.

## Figures and Tables

**Table 1 animals-09-00286-t001:** Land usage in four beef production systems.

Land Area (ha)	Extensive Unmodified Pasture	Feedlot Systems		Fertilised Irrigated Pasture	Semi-Intensive Silvopastoral Systems
		(a) Extensive Rearing	(b) Irrigated Pasture Rearing		
Land area (ha) occupied by cattle to produce 1 tonne of meat per annum (hot carcass weight)	27	17.1	5.2	10	2.2
Area for meat processing ha.tonne ^−1^ (hot carcass weight)	0.1	0.1	0.1	0.1	0.1
Area of maize etc. for feed ha.tonne ^−1^ (hot carcass weight)	0	3.6 ^1^	3.6 ^1^	0	0
Total land area ha.tonne ^−1^ (hot carcass weight)	27.1	20.8	8.9	10.1	2.3

^1^ The figure shown is for world average maize production. This figure would be 1.8 if the area needed for producing feed maize in the highest producing countries, like the USA, were used. The feedlot totals would then be 19.0 ha tonne ^−1^ for (a) and 7.1 ha tonne ^−1^ for (b).

**Table 2 animals-09-00286-t002:** Water usage in four beef production systems.

Water Usage (l·kg ^−1^ Hot Carcass Weight at 27 °C)	Extensive Unmodified Pasture	Feedlot Systems		Fertilised Irrigated Pasture	Semi-Intensive Silvopastoral Systems
		(a) Axtensive Rearing	(b) Irrigated Pasture Rearing		
Water drunk by animal	137	118.9 *	81 *	92	73.6
Evaporation from troughs	8.5	5 *	5 *	5	4.6
Washing troughs, cooling	<0.1	0.4 *	0.4 *	<0.1	<0.1
Loss of polluted water removed from troughs	<0.1	4 *	4 *	<0.1	<0.1
Meat processing water	9 *	9 *	9 *	9 *	9 *
Irrigation of crops: for feedlot animals	0	415 *	415 *	0	0
Irrigation of pasture	0	0	154 *	288 *	0
Water for fertilizer production (a) feed	0	0.2 *	0.2 *	0	0
Water for fertilizer production (b) on pasture	0	0	4.6 *	16.5 *	0
Total	154.5	552.5 *	673.2 *	410.5 *	87.2
Total: no irrigation of crops	154.5	137.5 *	258.2 *	410.5 *	87.2
Total: no irrigation of crops or pasture	154.5	137.5 *	104.2 *	122.5 *	87.2

* Some or all may be water that was purified for human use. (Totals not marked * if purified water used only for meat processing).

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
