# Peer review of "Land and Water Usage in Beef Production Systems"

_animals, 2019, doi:10.3390/ani9060286_

Round 1
Reviewer 1 Report
Summary
The paper reports the analysis of the land and water use of four beef production systems. This paper could serve as a potentially useful review for some readers. However, the novelty of the paper is low and the methodology applied is simple. For instance, others have already compared the water consumption of different cattle production systems and they have applied models which more greatly reflect the systems under consideration in my view (e.g. by taking into account livestock replacement rate, evapotranspiration etc.). But this simplicity may serve useful in future work, as long as the major assumptions made here are appreciated by the readers. Therefore, the author should take care to make these assumptions clear.
Broad comments
I think it needs to be clearer where the model is based. Obviously, the paper is not representative of Europe and it should be made clear that this is the case, as the title makes the paper sound more general than it is. The average temperature is 27C and, based on the references, the model is clearly most representative of South America. Thus, I feel that the title of the paper should be modified to reflect the fact it relates to subtropical conditions, or Latin America specifically. Furthermore, average rainfall should reflect this location and it was not clear to me if this was the case.
Overall, I feel the text would benefit from some editing because it is very difficult to read in its current state. Although well structured, I would appreciate more concise sentences throughout. Some additional attention is needed to make sure citations are all correct, as the style is inconsistent. I think the scope and functional unit should be more plainly outlined as well.
Specific comments
Line 9 – “negative in relation to resource use” what does this mean? Everything that requires resources could be classed as negative in relation to this – I think the author means compared to other meat production systems… please rephrase.
Line 26 to Line 29 – Please consider breaking this sentence up to make it more readable.
Line 42 – Very difficult to read so consider rephrasing to – “A system or procedure is sustainable if its effects are acceptable now and in the future in relation to resource availability, consequences of functioning and morality of action.”
Line 57 – “Every source of information is referenced.” As it should be, why does the author need to say this?
Line 83 – Again, consider breaking the sentence up, I am getting lost.
Line 92 – Rephrase – maybe remove “available in”
Line 113 to line 115 – The author gives two rearing durations and then says, “so these durations are used”. Which? Both? This should be made clear here. Was any sort of sensitivity performed, e.g. for the different rearing durations that were mentioned? If not, why?
Line 145 – “used in “the” calculations”.
Line 180 – Is land use for crop production only attributed to the space the crops occupy?
Line 218 – Citation style inconsistent.
Line 221 – “similar to those from several other studies.” Please give an example (e.g…..).
Line 250 – Too many “buts” and I lose the point.
Line 413 – the terms “blue water” and “green water” are suddenly introduced with no explanation as to what is meant here. Either add an explanation or (preferably) just explain using terminology already used in the paper.
Table 1 -Total land area ha.kg -1 unnecessary when ha.tonne -1 has been given.
Discussion – The author has touched upon the fact that land used for extensive production may not be practical for production of crops. I think this point needs to more greatly emphasised. The conclusion is that extensive rearing requires the most land, but this does not mean it is less sustainable. If crops would not be grown here anyway then the arable land use is actually lower for the extensive system.
Reviewer 2 Report
I've suggested some rephrasing throughout the article using Sticky notes in the attached document. I recommend reading some relevant meta-analyses papers on this subject and following their workflow, writing-style, and point out where your approach differs. Doing so would help the readability of this paper. Succinct writing could help with its readability. Although Figures are continuously mentioned they refer to Tables. I'm not sure the writing approach taken by this author falls within journal guidelines but I do know that I had a difficult time following. An Appendix filled with the various calculations used to generate the Results would make this study more reproducible.

Round 2
Reviewer 2 Report
The author has substantially improved the presentation and clarity of this paper.